

# Cellobiohydrolase B of *Aspergillus niger* over-expressed in *Pichia pastoris* stimulates hydrolysis of oil palm empty fruit bunches

James Sy-Keen Woon[1,2], Mukram M. Mackeen[3,4], Rosli M. Illias[5], Nor M. Mahadi[4,6], William J. Broughton[7], Abdul Munir Abdul Murad[1] and Farah Diba Abu Bakar[1]

[1] School of Biosciences and Biotechnology, Faculty of Science and Technology, Universiti Kebangsaan Malaysia, Bangi, Selangor, Malaysia
[2] Present address: Newcastle University Medicine Malaysia, Iskandar Puteri, Johor, Malaysia
[3] School of Chemical Sciences and Food Technology, Faculty of Science and Technology, Universiti Kebangsaan Malaysia, Bangi, Selangor, Malaysia
[4] Institute of Systems Biology (INBIOSIS), Universiti Kebangsaan Malaysia, Bangi, Selangor, Malaysia
[5] Department of Bioprocess Engineering, Faculty of Chemical Engineering, Universiti Teknologi Malaysia, Skudai, Johor, Malaysia
[6] Malaysia Genome Institute, Kajang, Selangor, Malaysia
[7] Department 4 (Materials & Environment), Federal Institute of Materials Research and Testing, Berlin, Germany

Corresponding author
Farah Diba Abu Bakar,
fabyff@ukm.edu.my

## ABSTRACT

**Background**. *Aspergillus niger*, along with many other lignocellulolytic fungi, has been widely used as a commercial workhorse for cellulase production. A fungal cellulase system generally includes three major classes of enzymes i.e., β-glucosidases, endoglucanases and cellobiohydrolases. Cellobiohydrolases (CBH) are vital to the degradation of crystalline cellulose present in lignocellulosic biomass. However, *A. niger* naturally secretes low levels of CBH. Hence, recombinant production of *A. niger* CBH is desirable to increase CBH production yield and also to allow biochemical characterisation of the recombinant CBH from *A. niger*.

**Methods**. In this study, the gene encoding a cellobiohydrolase B (*cbh*B) from *A. niger* ATCC 10574 was cloned and expressed in the methylotrophic yeast *Pichia pastoris* X-33. The recombinant CBHB was purified and characterised to study its biochemical and kinetic characteristics. To evaluate the potential of CBHB in assisting biomass conversion, CBHB was supplemented into a commercial cellulase preparation (Cellic® CTec2) and was used to hydrolyse oil palm empty fruit bunch (OPEFB), one of the most abundant lignocellulosic waste from the palm oil industry. To attain maximum saccharification, enzyme loadings were optimised by response surface methodology and the optimum point was validated experimentally. Hydrolysed OPEFB samples were analysed using attenuated total reflectance FTIR spectroscopy (ATR-FTIR) to screen for any compositional changes upon enzymatic treatment.

**Results**. Recombinant CBHB was over-expressed as a hyperglycosylated protein attached to *N*-glycans. CBHB was enzymatically active towards soluble substrates such as 4-methylumbelliferyl-β-D-cellobioside (MUC), *p*-nitrophenyl-cellobioside (*p*NPC) and *p*-nitrophenyl-cellobiotrioside (*p*NPG3) but was not active towards crystalline substrates like Avicel® and Sigmacell cellulose. Characterisation of purified CBHB

using MUC as the model substrate revealed that optimum catalysis occurred at 50 °C and pH 4 but the enzyme was stable between pH 3 to 10 and 30 to 80 °C. Although CBHB on its own was unable to digest crystalline substrates, supplementation of CBHB (0.37%) with Cellic® CTec2 (30%) increased saccharification of OPEFB by 27%. Compositional analyses of the treated OPEFB samples revealed that CBHB supplementation reduced peak intensities of both crystalline cellulose Iα and Iβ in the treated OPEFB samples.

**Discussion**. Since CBHB alone was inactive against crystalline cellulose, these data suggested that it might work synergistically with other components of Cellic® CTec2. CBHB supplements were desirable as they further increased hydrolysis of OPEFB when the performance of Cellic® CTec2 was theoretically capped at an enzyme loading of 34% in this study. Hence, *A. niger* CBHB was identified as a potential supplementary enzyme for the enzymatic hydrolysis of OPEFB.

# INTRODUCTION

Oil palm empty fruit bunches (OPEFB) are among the most abundant lignocellulosic biomass wastes of the palm oil industry. Research towards the utilisation of oil palm biomass into value-added products is desirable to better manage wastes and reduce carbon footprint as a result of OPEFB incineration. In addition, their renewability and low cost make OPEFB an ideal sugar-rich feed-stock for biofuel production (*Ibrahim et al., 2015*). Enzymatic hydrolysis of lignocellulosic biomass remains a bottleneck in biofuel production however due to the high cost and relative inefficiencies of lignocellulose degrading enzymes against recalcitrant biomass feed-stocks (*Brijwani, Oberoi & Vadlani, 2010*; *Fang et al., 2009*; *Klein-Marcuschamer et al., 2012*). *Aspergillus niger* is a filamentous fungus well-known for its ability to produce copious amount of lignocellulose degrading enzymes (*Saliu & Sani, 2012*). For this reason, numerous commercial enzyme preparations have been derived from *A. niger* cultures including Novozymes 188 (Novozymes Inc., Bagsvard, Denmark), BiocellulaseA (Quest Intl., Irvine, CA, USA) and Cellulase AP 30 K (Amano Enzyme Inc., Nagoya, Aichi, Japan) (*Verardi et al., 2012*). By nature, *A. niger* secretes large quantities of β-glucosidases but only limited amounts of cellobiohydrolases (CBH). Hence, characterisation of native *A. niger* CBHs has been underreported (*Dashtban, Schraft & Qin, 2009*; *Fang & Xia, 2015*). A related fungus, *Trichoderma reesei*, secretes large amounts of CBH but has very low extracellular β-glucosidase (BGL) activities because most of the activity is bound to the cell wall (*Bischof, Ramoni & Seiboth, 2016*). Therefore, enzyme supplements are often required for synergistic reasons to enhance the hydrolytic efficiencies of enzyme cocktails used on lignocellulosic feedstocks. Supplementation is also used because different types of biomass have different properties, therefore they require a set of enzymes that has to be tailor-made (*Stockton et al., 1991*; *Rosgaard et al., 2006*).

Cellobiohydrolases are essential components of the fungal cellulase system that include three major classes of enzymes i.e., β-glucosidases, endoglucanases and CBHs (*Garvey et al., 2013*; *Woon et al., 2016a*). CBHs are particularly important in biomass degradation due to their hydrolytic activities toward crystalline cellulose that is abundant in lignocellulose (*Abdeljabbar, Song & Link, 2012*). CBHs hydrolyse from the termini of cellulose chains in a processive manner, releasing cellobiose as the major hydrolysis product (*Teeri, 1997*). According to the glycoside hydrolase classification system, fungi generally produce two classes of CBHs (*Henrissat & Bairoch, 1996*). CBHs that belong to glycoside hydrolase (GH) family 6 (EC 3.2.1.176) are inverting enzymes that cleave from the reducing ends of cellulose chains whilst CBHs of GH family 7 (EC 3.2.1.91) are retaining enzymes that cleave from the non-reducing termini of cellulose chains (*Teeri, 1997*). Although heterologous expression of CBHs has been widely reported in yeasts (*Den Haan et al., 2013*) and fungi (*Zoglowek et al., 2015*), the use of recombinant cellobiohydrolases in the biomass conversion industry has been limited by their expression levels and unpredictable activity profiles caused by non-native glycosylation patterns of the expression hosts (*Gao et al., 2012*; *Jeoh et al., 2008*). Apart from glycosylation, CBH activity also depends on other posttranslational modifications such as N-terminal pyroglutamate formation that is commonly observed in crystal structures of Cel7A enzymes. This posttranslational modification however, is lacking in yeast expression systems (*Dana et al., 2014*). Low yields coupled with low enzymatic activities drive up the costs of enzymatic biomass conversion (*Klein-Marcuschamer et al., 2012*).

The methylotrophic yeast *Pichia pastoris* grows on simple media and to high densities in flasks and fermenters. It was chosen as the expression host since it also possesses many of the advantages of eukaryotic expression systems including protein processing, protein folding and posttranslational modifications (*Cregg, Vedvick & Raschke, 1993*). Accordingly, cellobiohydrolase B (CBHB) from *A. niger* ATCC 10574 was cloned and expressed in *P. pastoris* X-33. The enzymatic properties of purified CBHB were characterised and the recombinant CBHB was added to the commercial enzyme preparation (Cellic® CTec2) to investigate its synergistic effect on enzymatic hydrolysis of oil palm empty fruit bunches (OPEFB).

## MATERIALS AND METHODS

### Microbial strains and cloning vectors

*Escherichia coli* strain DH5 α (Promega, Madison, WI) was used for plasmid manipulation and propagation throughout as described by *Sambrook & Russell (2001)*. The previously isolated cDNA of *cbh*B (GenBank accession number: KR052992.1) was cloned into the pGEM-T Easy cloning vector (Promega) (*Woon, Murad & Abu Bakar, 2015*). *P. pastoris* strain X-33 (Invitrogen/Life Technologies, Grand Island, NY) was used for the expression of *A. niger cbh* B. Preparation of media and culturing of yeast transformant were carried out following Invitrogen/Life Technologies *Pichia* expression system protocols.

## Construction of the expression cassette

The cloning vector harbouring the cDNA of *cbh*B was extracted from *E. coli* DH5 α cultured in 10 mL Luria-Bertani medium using the Wizard® *Plus* SV Minipreps DNA Purification System (Promega). To construct an expression cassette of the target gene, the plasmid from *E. coli* was digested with *Cla*I and *Kpn*I (New England Biolabs, Beverly, MA) to release the insert, followed by gel purification of the target gene using the MEGAquick-spin™ Total DNA Fragment Purification kit (*INTRON* Biotechnology, Gyeonggi-do, South Korea) before it was inserted into the expression vector pPICzαC (Invitrogen/Life Technologies). Manipulation of DNA was carried out using standard procedures (*Sambrook & Russell, 2001*).

## Transformation of *P. pastoris* and screening of transformants

Transformation of *P. pastoris* X-33 was performed according to the Invitrogen/Life Technologies *Pichia* expression system protocol. Transformants containing multiple insertions were selected on yeast extract peptone dextrose plates containing sorbitol (YPDS) and amended with various concentrations of Zeocin™ (500, 1,000, and 2,000 $\mu$g mL$^{-1}$). Positive transformants were confirmed by PCR (*Lõoke, Kristjuhan & Kristjuhan, 2011*) using specific primers (5′-*cbh*B forward: ATCGATGCATCATCATCATCATCATCAGCAGGTT and 3′-*cbh*B reverse: CCGGTACCTCACAAACACTGCGAGTA) to detect the target gene within the *P. pastoris* genome.

## Production and purification of CBHB

Transformants carrying the expression construct *cbh* B_pPICZαC were inoculated into 100 mL of buffered glycerol complex medium and cultured to an OD$_{600}$ of 2 to 3 at 30 °C with shaking at 240 rpm. Then the cells were harvested via centrifugation ($1,500 \times$ g, 5 min) and re-suspended in 50 mL of buffered methanol complex medium. Methanol was added every 24 h to a final concentration of 1.0% (v/v) during the 72 h incubation (at 30 °C with shaking at 240 rpm). Culture supernatants were clarified by centrifugation ($3,000 \times$ g, 5 min) and stored at $-20$ °C. CBHB production was verified by SDS-PAGE (12% polyacrylamide) followed by western-blot analyses using mouse anti His-tag monoclonal antibodies (Cat# 70796, Novagen, Madison, USA) and HRP-conjugated anti-mouse antibodies (Promega) for chemi-luminescent detection on X-ray films. Protein concentrations were determined using the Bradford method (*Bradford, 1976*).

Culture supernatants containing His$_6$-tagged CBHB recombinant enzymes were purified by immobilised metal-ion affinity chromatography (AKTA prime system from GE Healthcare Bio-Sciences Corp., NJ, USA) using a 1 mL HiTrap chelating column charged with Ni$^{2+}$ ions. The column was equilibrated with 10 mL binding buffer (50 mM NaH$_2$PO$_4$ (pH 8.0), 0.5 M NaCl, 20 mM imidazole). Crude protein (1 mL) was loaded onto the column and the resin washed with 10 mL binding buffer. Bound protein was eluted by a linear gradient of elution buffer (50 mM NaH$_2$PO$_4$ (pH 8.0), 0.5 mM NaCl, 50 to 400 mM imidazole). Eluted fractions that contained high concentrations of proteins were pooled and concentrated using Vivaspin™ centrifugal concentrators (cut-off of 10 kDa; GE Healthcare Bio-Sciences Corp.).

To assess the extent of $N$-glycosylation of CBHB, 20 µg of purified CBHB was deglycosylated using PNGaseF (New England Biolabs, Beverly, MA, USA) according to the manufacturer's instructions. Deglycosylated CBHB was analysed by SDS-PAGE to compare the molecular weight of the enzyme before and after deglycosylation.

### Enzyme assays

The catalytic activity (initial rates) of CBHB was measured using 4-methylumbelliferyl-β-D-cellobioside (MUC) (Sigma–Aldrich Corp., St. Louis, MO, USA) in a 400 µL reaction mixture containing 30 mM citrate buffer (pH 4.0), 0.4 mM MUC and 3 µg of purified CBHB. After 15 min incubation at 50 °C, reactions were terminated by adding 100 µL of 1.0 M $Na_2CO_3$. Fluorescence of the 4-methylumbelliferyl group released from MUC was determined at excitation and emission wavelengths of 365 and 460 nm, respectively. One unit of enzyme activity was defined as the amount of the enzyme that produced the equivalent of 1 µmol product (4-methylumbelliferrone) under optimal conditions in 1 min. Enzyme assays were also performed using $p$-nitrophenyl-β-D-cellobioside ($p$NPC), $p$-nitrophenyl-β-D-cellobiotrioside ($p$NPG3) and $p$-nitrophenyl-β-D-lactoside ($p$NPL) as substrates under similar assay conditions. Optical densities were read at 405 nm to detect the hydrolysis product ($p$-nitrophenol). To investigate CBHB activity towards crystalline substrates, assays were performed in 400 µL of 30 mM citrate buffer (pH 4.0) using 1% (w/v) microcrystalline cellulose Avicel® PH-101 (Sigma–Aldrich) or 1% (w/v) Sigmacell Cellulose Type 20 (Sigma–Aldrich) and 3 µg of purified CBHB. Assay mixtures were incubated at 50 °C for 30 min; reactions were then terminated by boiling for 10 min. The amount of reducing sugar produced was estimated using the dinitrosalicylic (DNS) reagent method (*Miller, 1959*).

### Thermal and pH profiles of CBHB

Purified CBHB (3 µg) was incubated at various temperatures (30, 40, 45, 50, 55, 60, 70 and 80 °C) in 250 µL citrate buffer (pH 4.0, 30 mM) for 30 min. Upon cooling down on ice for 1 min, MUC (at 0.4 mM final concentration) and deionized distilled water were added to top up the reaction mixtures to a final volume of 400 µL. Residual activities of the treated protein was assayed using the standard protocol described above and MUC as the substrate.

Purified CBHB (3 µg) was also incubated at various pH buffers (pH 3–10) for 30 min. Subsequently, MUC (at 0.4 mM final concentration) and deionised distilled water were added to top up the reaction mixtures to a final volume of 400 µL. Residual activities of the treated protein was assayed using the standard protocol described above and MUC as the substrate.

### Enzyme kinetics analysis

Michaelis–Menten kinetics of purified CBHB were determined under optimal catalytic conditions for purified CBHB (3 µg or 7.5 µg $mL^{-1}$) i.e., 50 °C and pH 4.0 in a 30 mM citrate buffer. Multiple concentrations of MUC ranging from 0.5 to 2.0 mM were used to gather data on enzyme kinetics. A Lineweaver-Burk plot (*Lineweaver & Burk, 1934*) was used to calculate the values of $K_m$ and $V_{max}$ of CBHB.

## Effect of metal-ions and reagents on enzyme activity

Reaction mixtures consisted of 3 µg enzyme, 250 µL citrate buffer (pH 4.0, final concentration—30 mM), 100 µL MUC (final concentration—0.4 mM) and 50 µL solutions containing different metal ions ($Ba^{2+}$, $Ca^{2+}$, $Co^{2+}$, $Cu^{2+}$, $Fe^{2+}$, $K^+$, $Mg^{2+}$, $Mn^{2+}$, $Na^+$ and $Zn^{2+}$) at final concentrations of 1 and 10 mM, respectively. The effects of EDTA (1 and 10 mM) and urea (0.1 and 1.0 M) on enzyme activity under standard conditions were also tested.

## Enzymatic hydrolysis of OPEFB using Cellic®CTec2 supplemented with CBHB

Sime Darby Limited (Pulau Carey, Selangor, Malaysia) kindly provided and pre-treated the OPEFB samples as follows: OPEFB samples were dried at 60 °C to constant mass and ground to pass a 0.25 mm sieve. To remove surface lignin, 1 g ground OPEFB was treated with 8 mL of calcium hydroxide (1 g $L^{-1}$) for 90 min at 50 °C. Samples were then washed three times with deionised distilled water and recovered by filtration. Then, 20% (v/v) of peracetic acid was added at a v/w ratio of 1:1 ratio and incubated for 2 h at 75 °C. Peracetic acid was prepared earlier by mixing 600 mL of glacial acetic acid with 400 mL of 30% hydrogen peroxide at room temperature for 72 h. Pre-treated OPEFB samples were then washed with deionised distilled water until neutral pH, then dried to constant mass at 60 °C.

To investigate the synergistic effect of CBHB on hydrolysis of OPEFB, purified CBHB (10–50 µg) was supplemented to the commercial enzyme cocktail Cellic® CTec2. Saccharification of OPEFB was performed in 1 mL reaction mixtures containing 25 mM pH 5 sodium acetate buffer and 1% (w/v) pre-treated OPEFB at 50 °C with shaking at 1,000 rpm for 5 h (Thermomixer Comfort, Eppendorf AG, Hamburg, Germany). Production of reducing sugars were measured using the DNS reagent (*Miller, 1959*) and total reducing sugars (TRS) calculated using the following equation (*Farias Silva et al., 2015*):

$$\%\mathrm{TRS} = 100 \times \left( \frac{C \times V}{W} \right) \tag{1}$$

Where: c (mg $mL^{-1}$) = concentration of reducing sugars as measured by the DNS assay;

$v$ (mL) = volume of the OPEFB hydrolysis reaction mixture;

$w$ (mg) = weight of pre-treated OPEFB used in hydrolysis assay.

To achieve maximum saccharification of OPEFB, enzyme loadings were optimised using response surface methods (RSM) and a central composite rotatable design (CCRD) to determine the effects of independent variables on enzymatic hydrolysis. Thirteen experimental runs were formed by Design Expert Version 6.0.10 (Stat Ease Inc., Minneapolis, MN, USA) with five replications at the central points, four replications at the axial points and four replications at the factorial points. The variables were: Cellic®CTec2 loadings (A) and CBHB loadings (B). Models were deemed suitable when they were significant based on ANOVA ($P < 0.05$) and insignificant based on lack-of-fit tests ($P > 0.05$) (*Whitcomb & Anderson, 2004*).

The predicted optimum point was confirmed experimentally in three replications and was validated using Root Mean Squared Deviation (RMSD) as described by *Pineiro et al. (2008)*:

$$\text{RMSD} = \sqrt{\frac{1}{n-1} \sum_{i=n} (\hat{y}_i - y_i)^2} \qquad (2)$$

where $\hat{y}_i =$ observed value; $y_i =$ predicted value; $n =$ number of replicates.

Treated OPEFB was separated from the supernatant by centrifugation (3,000× g, 5 min) and dried on a hotplate. Dried OPEFB samples were analysed using a Perkin Elmer (Waltham, MA, USA) Spectrum 400 Series Fourier transform infrared spectroscopy instrument fitted with a universal ATR sampler accessory (ATR-FTIR spectroscopy). The scanning range of the experiment was 650 to 4,000 cm$^{-1}$.

# RESULTS AND DISCUSSION

## Construction of an expression cassette and transformation of *P. pastoris*

Transformation of *P. pastoris* with the expression cassette yielded >30 transformants, 16 of which were randomly picked and re-streaked on YPDS plates containing 500 µg mL$^{-1}$ Zeocin$^{TM}$. All grew well but upon transfer to 1,000 µg mL$^{-1}$ Zeocin$^{TM}$, the number of survivors declined to 13 while only eight colonies withstood 2,000 µg mL$^{-1}$ Zeocin$^{TM}$. These different survival abilities reflected the number of integrations of the expression cassette into the genome of *P. pastoris*. Generally, transformants that are tolerant to higher Zeocin$^{TM}$ levels are expected to harbour more copies of the gene and hence, higher expression levels of the target gene (*Nordén et al., 2011*). In this study however, we found that a transformant (K2) tolerant to 1,000 µg mL$^{-1}$ Zeocin$^{TM}$ produced the highest amounts of crude proteins (6.5 mg mL$^{-1}$) amongst all selected transformants with different Zeocin$^{TM}$ tolerances. Selected transformants were also screened by colony PCR (*Lõoke, Kristjuhan & Kristjuhan, 2011*) whereby all positive transformants yielded amplicons of ~1.6 kb equivalent to the full-length size of *cbh*B (Fig. 1).

## Production and purification of CBHB

CBHB was over-expressed in the X-33 transformant relative to the host proteins produced by the untransformed X-33 strain (Fig. 2). The recombinant CBHB was purified to apparent homogeneity via IMAC purification and its identity confirmed by western blotting (Fig. 3). Table 1 summarises the purification steps used for CBHB. As much as 0.01 mg mL$^{-1}$ (mg enzyme/mL culture) of purified CBHB was recovered following ultracentrifugation and purification by IMAC. Judging from the very low yield of the IMAC-purified sample (9%) (Table 1), there was a significant loss of enzymatic activity caused either by protein degradation or protein loss during the purification process.

According to *Den Haan et al. (2013)*, production titres of recombinant CBH in yeasts such as *P. pastoris*, *Saccharomyces cerevisiae* and *Yarrowia lipolytica* varied vastly from 0.0001 mg mL$^{-1}$ to ~1.7 mg mL$^{-1}$. In most cases however, expression of CBH remained very low ranging from 0.0001 to 0.01 mg mL$^{-1}$. *Li et al. (2009)* reported the highest level

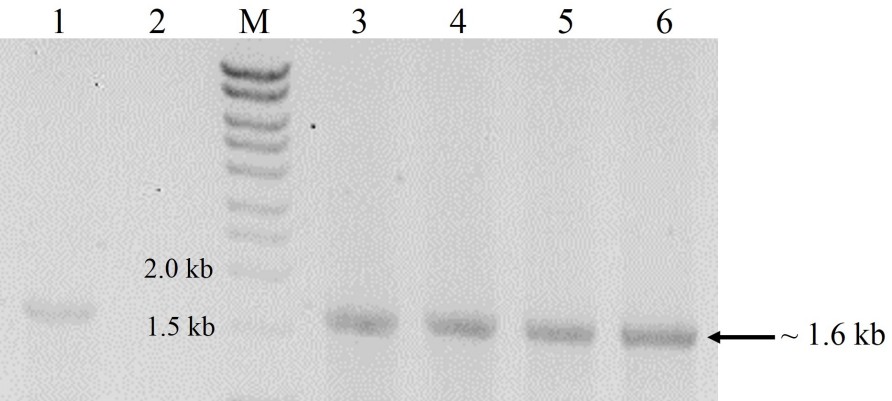

**Figure 1** **Agarose gel electrophoresis profiles of DNA extracted from *P. pastoris* X-33 transformants.** Lane 1, Positive PCR control (expression cassette *cbh*B_pPICZαC used as DNA template); Lane 2, Negative PCR control (without DNA template); Lane M, molecular weight marker; Lanes 3 to 6, DNA amplified from colonies of *P. pastoris* X-33 transformed with *cbh*B (K1-K4).

**Table 1** **Purification of *A. niger* CBHB expressed in *P. pastoris* X-33.**

| Purification steps | Volume (mL) | Activity (U mL⁻¹) | Total activity (U)[a] | Total protein (mg) | Specific activity (U mg⁻¹) | Purification fold[b] | Yield[c] (%) |
|---|---|---|---|---|---|---|---|
| Crude protein | 100 | 0.02 | 2.00 | 20.8 | 0.096 | 1 | 100 |
| Ultracentrifugation | 4 | 0.20 | 0.80 | 3.96 | 0.201 | 2.1 | 40 |
| IMAC | 1 | 0.18 | 0.18 | 0.99 | 0.182 | 1.9 | 9 |

**Notes.**
[a]U = Amount of enzyme required to produce 1 μmol of methylumbeliferone per min under specific conditions.
[b]Purification fold = Specific activity of a purified sample divided by the specific activity of the crude protein.
[c]Yield = Total activity a purified sample divided by the total activity of the crude protein.

of recombinant CBH expression in yeast to date. A thermostable CBH from *Chaetomium thermophilum* was expressed in *P. pastoris* at a titre of ∼1.7 mg mL⁻¹. Whereas in this study, the *P. pastoris* transformant carrying *cbh*B had an enzymatic activity of 0.02 U mL⁻¹ and the expression level was 0.21 mg mL⁻¹ after 3 d induction (Table 1).

According to the ProtParam server, the calculated molecular mass of CBHB is ∼64 kDa (http://web.expasy.org/protparam/). However, purified CBHB displayed an apparent molecular weight of >100 kDa on SDS-PAGE (Fig. 4). *In silico* analyses of CBHB amino acid sequences using the NetNGlyC 1.0 server (http://www.cbs.dtu.dk/services/NetNGlyc/) revealed the presence of two putative *N*-glycosylation sites on CBHB (N̲GS and N̲SS, at the 308th and 371th amino acid residues, respectively). De-glycosylation of CBHB using PNGaseF confirmed the presence of the *N*-glycans experimentally. Fig. 4 clearly shows that removal of *N*-glycans by PNGaseF reduced the molecular mass of CBHB to a size between 70 and 100 kDa. Besides, CBHB might also be *O*-glycosylated as it contained the Pro/Ser/Thr-rich linker peptide (residues 459–500th, GenBank accession number: KR052992.1) that is prone to *O*-glycosylations (*Beckham et al., 2012*). As such, it is prudent to conclude that CBHB was produced as a hyper-glycosylated protein in *P. pastoris* X-33.

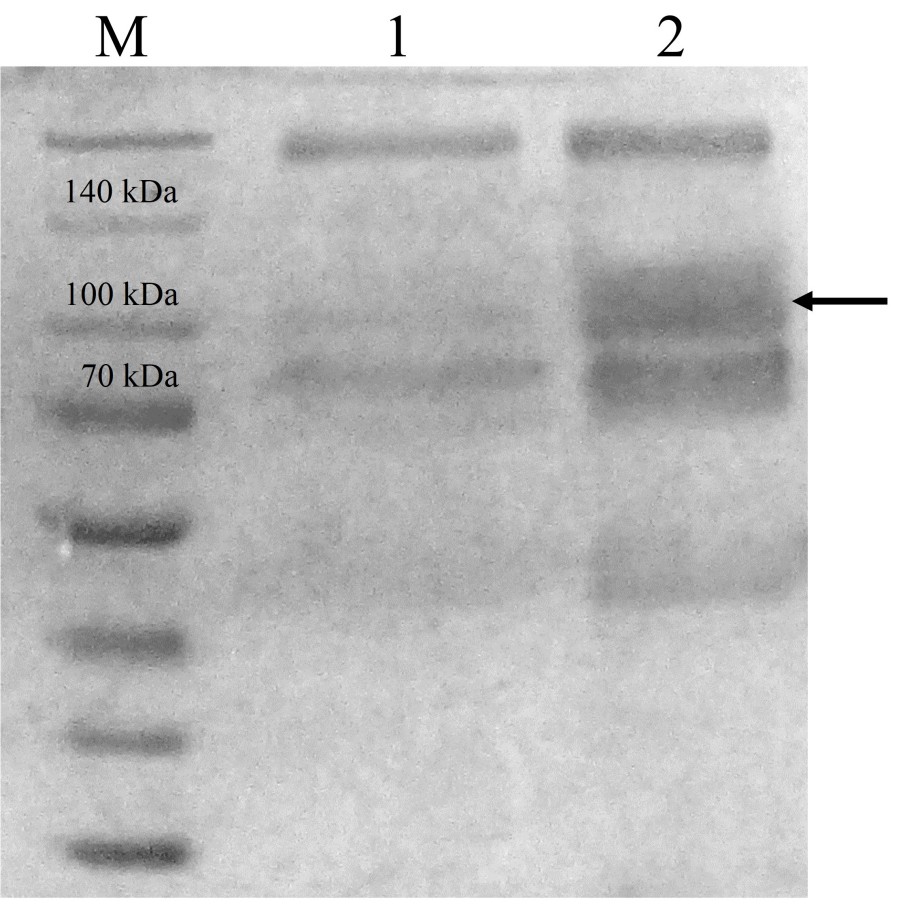

**Figure 2** **Sodium dodecyl sulfate-polyacrylamide gel electrophoresis (SDS-PAGE) profiles of concentrated crude protein extracts from *P. pastoris* X-33.** Lane M, molecular weight marker; Lane 1, concentrated crude proteins from untransformed *P. pastoris* X-33; Lane 2, concentrated crude proteins from a *P. pastoris* X-33 transformant (K2) harbouring the expression cassette *cbh*B_pPICZαC. The black arrow indicates recombinant CbhB.

Glycosylations of CBHs could unpredictably interfere with enzyme activities, stabilities and substrate bindings. The extent and heterogeneity of glycosylations are dependent on many factors such as growth conditions, expression host and the presence of glycan trimming enzymes in the secretome (*Beckham et al., 2012*). As the effects of glycosylation on CBHs are always case-dependent, a full enzyme characterisation is required to investigate its enzymatic properties and potential usefulness.

### Enzymatic properties of CBHB

Optimal catalysis of CBHB with 4-methylumbelliferyl-β-D-cellobioside (MUC) occurred at 50 °C and pH 4 (Figs. 5A and 5B). The enzyme was stable across a wide range of pH (pH 3 to 10) and retained more than 80% of its activity after incubation for 30 min under different pH conditions (Fig. 5D). As CBHB originated from the mesophilic fungus *A. niger*, it was relatively thermo-tolerant. CBHB retained more than 80% of its activity after incubation at 30 to 80 °C for 30 min (Fig. 5C). To investigate long-term stability at its

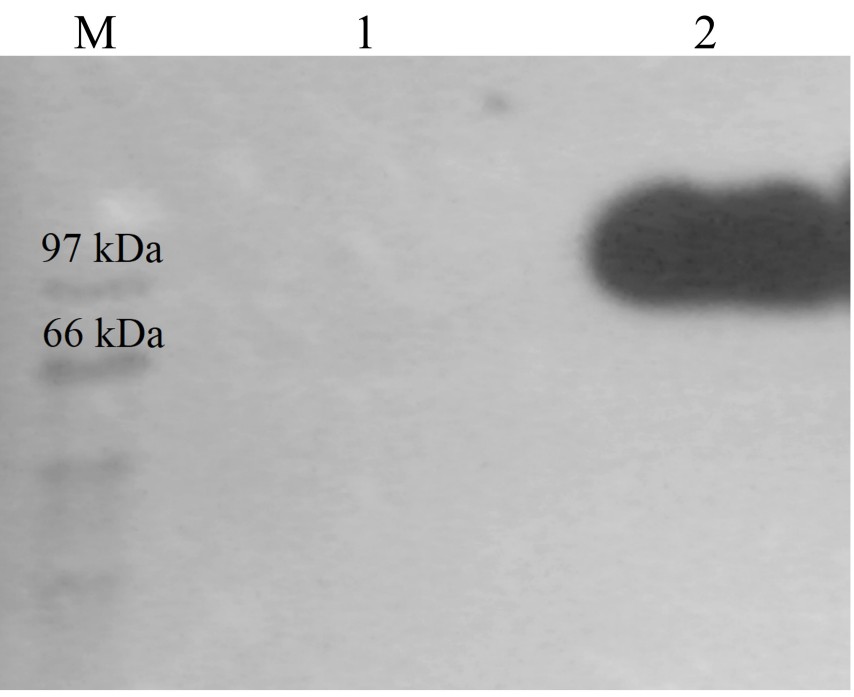

**Figure 3** **Western detection of purified CbhB from *P. pastoris* X-33.** Lane M, molecular weight marker; Lane 1, Negative control (concentrated crude proteins from untransformed *P. pastoris* X-33); Lane 2, purified CbhB.

optimum temperature, CBHB was incubated at 50 °C for 1 to 5 h prior to enzyme assays. Under these conditions, CBHB had a half-life of 2 h and retained ∼30% of its activity after 5 h of incubation. Stability at 50 °C is desirable as most commercial cellulase preparations optimally catalyse at the same temperature (*Verardi et al., 2012*).

CBHB was also active towards *p*-nitrophenyl-cellobioside (*p*NPC) and *p*-nitrophenyl-cellobiotrioside (*p*NPG3) (Table 2). Both *p*NPC and *p*NPG3 are synthetic substrates conjugated at their reducing termini with chromogenic *p*-nitrophenol. In contrast, enzymatic activity was not detected with *p*-nitrophenyl-lactoside (*p*NPL) another commonly used substrate for CBH detection (*Godbole et al., 1999*). As expected CBHB had no effect on carboxymethylcellulose (CMC), an amorphous cellulose that is generally used in endoglucanase assays (*Quay et al., 2011*). Interestingly however, a recombinant CBH1 from *A. niger* expressed in *P. pastoris* KM71H was active towards CMC (*Li et al., 2012*). In this study, CBHB was unable to hydrolyse the microcrystalline substrates Avicel® and Sigmacell cellulose. Highly crystalline Avicel® is commonly used in CBH assays and it has been so widely used that some CBHs are also known as avicelases (*Bronnenmeier, Rücknagel & Staudenbauer, 1991*).

Other reports of CBHs derived from *P. pastoris* mentioned decreased activity towards insoluble crystalline substrates such as Avicel® or bacterial microcrystalline cellulose (BMCC) (*Boer, Teeri & Koivula, 2000*; *Bronnenmeier, Rücknagel & Staudenbauer, 1991*; *Kanokratana et al., 2008*; *Woon et al., 2017*). Most probably, this decreased activity is linked
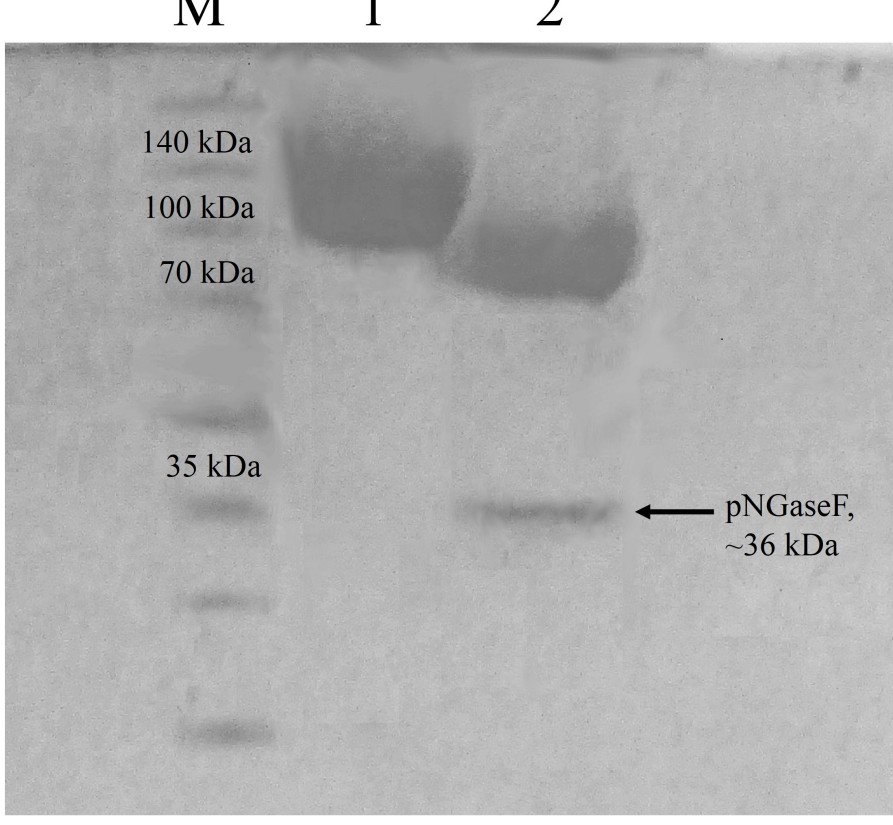

**Figure 4** **Sodium dodecyl sulfate-polyacrylamide gel electrophoresis (SDS-PAGE) profiles of purified CbhB.** Lane M, molecular weight marker; Lane 1, IMAC purified CbhB; Lane 2, IMAC- purified CBHB deglycosylated using PNGaseF.

**Table 2** **Substrate specificities of purified CBHB assayed at 50 °C, pH 4.**

| Substrate | MUC | *p*NPC | *p*NPG3 | *p*NPL | CMC | Avicel® | Sigmacell cellulose |
|---|---|---|---|---|---|---|---|
| Specific activity (U mg$^{-1}$) | 0.19 | 0.15 | 0.11 | N.D[a] | N.D[b] | N.D[b] | N.D[b] |

**Notes.**

MUC, *4*- methylumbelliferylβ-D-cellobioside; *p*NPC, *p*-nitrophenol-β-D-cellobioside; *p*NPG3, *p*-nitrophenol-β-D-cellotrioside; *p*NPL, *p*-nitrophenol-β-D-lactoside; CMC, carboxymethylcellulose; N.D, not detected within the concentration range of the standard chemical compounds used in the calibration curves.
[a]The range of *p*NP concentrations used in the calibration curve was 0.03–0.21 μmol.
[b]The range of glucose concentrations used in the DNS calibration curve was 0.25–8.0 μmol.

to the high glycan content (both *N*- and *O*-linked) of CBHs that potentially perturbs folding of the enzyme, substrate binding and enzyme activity (*Gao et al., 2012*; *Woon et al., 2016b*). The $K_m$ and $V_{max}$ of purified CBHB were calculated to be 0.25 mM and 1.41 U mg$^{-1}$, respectively using MUC as the substrate (Fig. 6). The turnover number ($k_{cat}$) was 2.36 s$^{-1}$.

To test the effect of metal-ions, CBHB was incubated with various salts and assayed for cellobiohydrolase activity against MUC. None of the ions activated the enzyme but at 10 mM, $Fe^{3+}$ ions inhibited activity by ~90% (Fig. 7). Inhibition of cellulase activity by

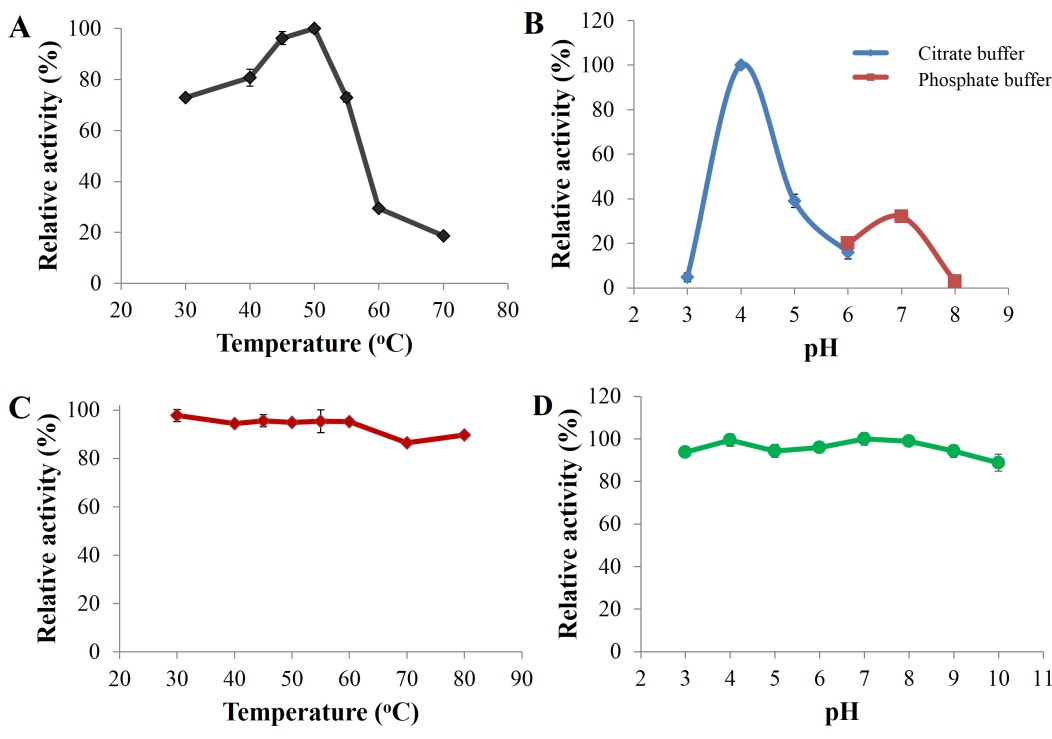

**Figure 5** **Enzyme activity profiles of CbhB.** (A) CbhB activity profile over a range of temperatures displaying the optimum temperature, (B) CbhB activity profile over a range of pH showing the optimum pH, (C) CbhB activity profile showing thermostability after exposure to different temperatures (30, 40, 45, 50, 55, 60, 70, 80 °C) for 30 min and (D) CbhB activity profile displaying pH stability after exposure to various pH (pH 3 to 10) for 30 min. All assays were performed using 4-methylumbelliferyl β-D-cellobioside (MUC) as the substrate.

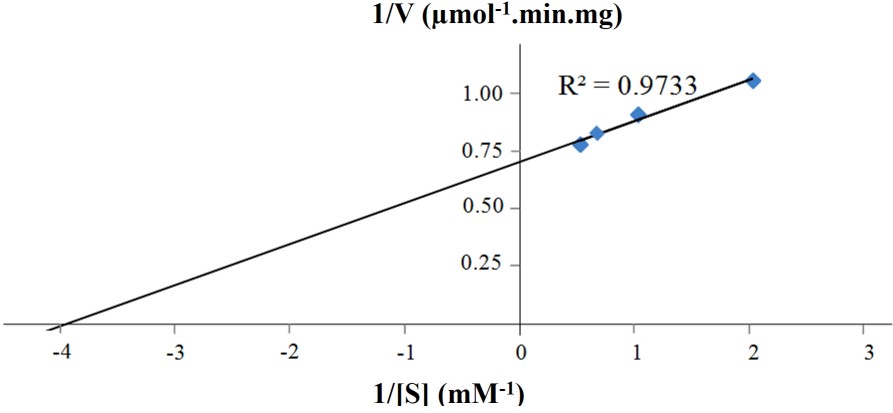

**Figure 6** **Lineweaver-Burk plot of purified CbhB assayed at 50 °C and pH 4 using different 4-methylumbelliferyl-β-D-cellobioside (MUC) substrate concentrations (0.5 mM, 1.0 mM, 1.5 mM and 2.0 mM).**

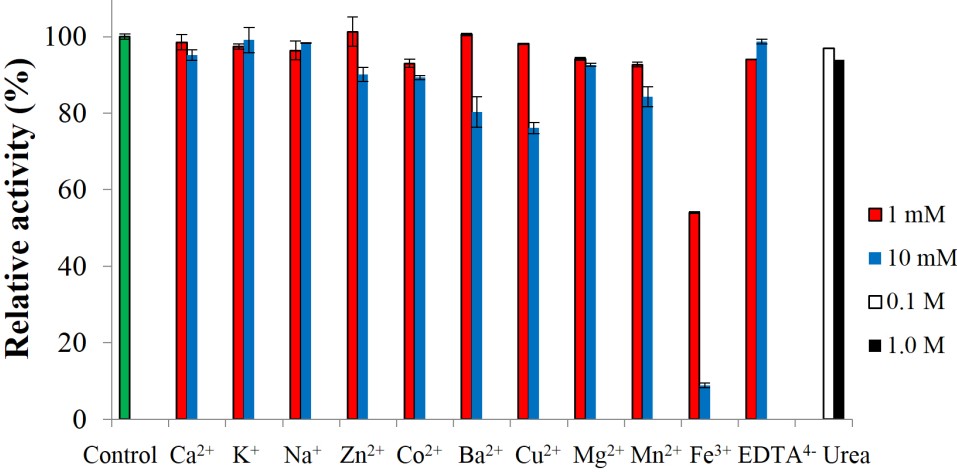

**Figure 7** Effect of metal-ions (1 mM and 10 mM), ethylenediaminetetraacetic acid (EDTA) (1 mM and 10 mM) and urea (0.1 M and 1.0 M) on the activity of CbhB.

$Fe^{3+}$ ions was also reported by *Wang et al. (2012)* who suggested that $Fe^{3+}$ ions lowered the digestibility of cellulose substrates via oxidation of reducing termini. Inhibition by metal-ions can also be reduced by supplementing reaction mixtures with chelating agents such as ethylenediaminetetraacetic acid (EDTA) (*Tejirian & Xu, 2010*). Addition of other metal-ions and reagents ($Ca^{2+}$, $K^+$, $Na^+$, $Zn^{2+}$, $Co^{2+}$, $Mg^{2+}$, $Mn^{2+}$ $EDTA^{4-}$ and urea) did not significantly interfere with CBHB activity. At 10 mM however, the presence of $Ba^{2+}$ and $Cu^{2+}$ reduced catalytic activity by 20–25%. Insensitivity of CBHB towards metal ions and denaturing reagents is a desirable trait in harsh industrial applications.

Attempts to produce *A. niger* CBHB in *Escherichia coli* Origami™ DE3 failed because the enzyme was localised in insoluble bodies that pelleted with the bacterial cells (*Woon, Murad & Abu Bakar, 2015*). *Gurgu & Barbu (2013)* attempted to express CBHB in *S. cerevisiae* BY4741 (*MATα his3 leu2 met15 ura3*) but the enzyme was not active against Avicel®. Interestingly, expression of CBHB in *S. cerevisiae* Y294 (*α leu2-3112 ura3-52 his3 trp1-289*) attempted by *Den Haan et al. (2007)* produced a functional enzyme active towards BMCC, albeit at a very low specific activity (0.03 U mg$^{-1}$). In our work, CBHB of *A. niger* expressed in *P. pastoris* X-33 produced an enzyme that was active against short-chain glycosides but not against crystalline substrates.

## Enzymatic hydrolysis of OPEFB using Cellic® CTec2 supplemented with CBHB

The enzyme loading was optimised at different concentrations of Cellic® CTec2 (1.5 to 30%) and CBHB (0.1 to 0.5%) (Table 3). The equation that best described enzymatic saccharification of OPEFB was sought by fitting various mathematical models including linear, quadratic and 2FI to the data. As the *P* value of ANOVA was lower than 0.05 (0.034) and the lack of fit test was insignificant (*P* value = 0.24) (Table 4), quadratic model best fitted the experimental data. The $R^2$ value for this model was 0.93, showing that only 7% of

**Table 3** Experimental design and results of CCRD for enzymatic hydrolysis of pre-treated oil palm empty fruit bunches (OPEFB).

| Run number | A: Cellic® CTec2 (% w/w)[a] | B: CBHB (% w/w)[a] | Reducing sugar production (mM) | |
|---|---|---|---|---|
| | | | Experimental | Predicted |
| 11 | 15.3 | 0.3 | 18.8 | 18.0 |
| 12 | 15.3 | 0.3 | 18.6 | 18.0 |
| 1 | 5.5 | 0.1 | 13.0 | 11.5 |
| 6 | 29.0 | 0.3 | 21.3 | 21.1 |
| 4 | 25.0 | 0.5 | 19.7 | 20.2 |
| 2 | 25.0 | 0.1 | 19.4 | 18.7 |
| 5 | 1.5 | 0.3 | 9.5 | 10.5 |
| 13 | 15.3 | 0.3 | 18.4 | 18.0 |
| 8 | 15.3 | 0.6 | 16.8 | 16.4 |
| 9 | 15.3 | 0.3 | 18.0 | 18.0 |
| 3 | 5.5 | 0.5 | 12.7 | 12.4 |
| 7 | 15.3 | 0.0 | 13.5 | 14.8 |
| 10 | 15.3 | 0.3 | 16.4 | 18.0 |

**Notes.**
[a] (% w/w) = (mg enzyme/mg pre-treated OPEFB) × 100%.

**Table 4** ANOVA for response surface models that describe the enzymatic saccharification of pre-treated oil palm empty fruit bunch (OPEFB).

| Source | Sum of squares | Degree of freedom | Mean square | F value | P value Prob > F | $R^2$ |
|---|---|---|---|---|---|---|
| Sequential model sum of squares | | | | | | |
| Linear | 116 | 2 | 57.8 | 22.1 | 0.0002 | 0.82 |
| 2FI | 0.08 | 1 | 0.08 | 0.03 | 0.8685 | 0.82 |
| Quadratic | 16.2 | 2 | 8.08 | 5.69 | 0.0340 | 0.93 |
| Cubic | 4.21 | 2 | 2.11 | 1.8400 | 0.96 | |
| Lack of Fit tests | | | | | | |
| Linear | 22.4 | 6 | 3.73 | 3.92 | 0.1034 | – |
| 2FI | 22.3 | 5 | 4.46 | 4.69 | 0.0797 | – |
| Quadratic | 6.1 | 3 | 2.04 | 2.15 | 0.2365 | – |
| Cubic | 1.92 | 1 | 1.92 | 2.02 | 0.2286 | – |
| Pure error | 3.80 | 4 | 0.95 | | | – |

the variation in response could not be explained by this model. In terms of decoded values the equation was:

$$Y = 6.25 + 0.71A + 19.85B - 0.0114A^2 - 30.08B^2 + 0.07AB, \tag{3}$$

where $Y$ = production of reducing sugar (mM); A = enzyme loading of Cellic® CTec2 (1.5–30%); B = enzyme loading of CBHB (0.1–0.5%).

The response-surface diagram indicated that saccharification of OPEFB increased with the concentration of Cellic® CTec2 (Fig. 8). At a Cellic® CTec2 concentration of >25%, production of reducing sugars was less affected by further increases in Cellic® CTec2 levels.
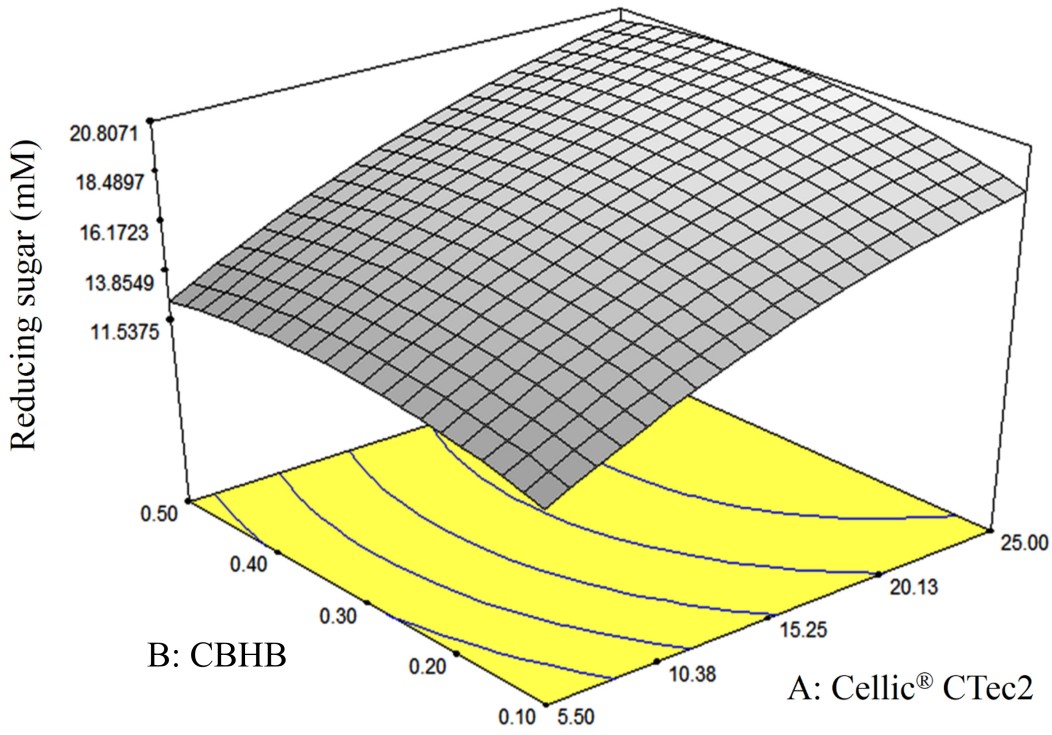

**Figure 8** **Response surface diagrams of oil palm empty fruit bunch (OPEFB) hydrolyses using various loadings of Cellic® CTec2 (5.5–25%) and CBHB (0.1–0.5%).** Production of reducing sugars was measured using DNS assays (Section 2.5). The contour lines are the two dimensional representation of the response surface. All points within the same contour line have equal reducing sugar concentration.

Maximum saccharification of OPEFB was predicted to occur when 30% Cellic® CTec2 and 0.37% CBHB were to use in OPEFB hydrolysis.

To validate these predicted saccharification optima, an experiment was conducted (in triplicate) using the optimised enzyme loadings. Total reducing sugars (TRS) produced by the optimised mixture amounted to 38.8% which was in good agreement with the 38.6% predicted by the RSM model (Table 5, (c)). Values observed at the optima were subjected to Root Mean Squared Deviation (RMSD) calculations to determine the accuracy of the RSM model. As a rule of thumb, the RSM model is deemed accurate when the value of RMSD is smaller than 10% of the predicted value (*Whitcomb & Anderson, 2004*). In our work, the RMSD value at the optima of 1.57 (which is <10% of 21.4 mM, Table 5, (c)) indicated that the RSM model accurately predicted optimum enzyme loadings for maximum saccharification.

As mentioned above, saccharification of OPEFB peaked when the concentration of Cellic® CTec2 loaded was more than 25%. To calculate the concentration of Cellic® CTec2 where OPEFB saccharification is maximised, another Eq. (3) was derived that only considered Cellic® CTec2 (A):

$$dY/dA = 0.71 - 0.0228A, \tag{4}$$

where: A = enzyme loading of Cellic® CTec2 (1.5 to 30%).

**Table 5  Enzymatic hydrolysis of oil palm empty fruit bunch (OPEFB) by Cellic® CTec2 and CBHB.**

| Samples | Reducing sugar (mM) | Total reducing sugars (%)[c] |
|---|---|---|
| (a) OPEFB hydrolysed by CBHB[a] | 0 | 0 |
| (b) OPEFB hydrolysed by Cellic® CTec2[b] | $16.9 \pm 0.3$ | $30.5 \pm 0.6$ |
| (c) OPEFB hydrolysed by Cellic® CTec2[b] and CBHB[a] | $21.5 \pm 1.3$ (21.4)[d] | $38.8 \pm 2.4$ (38.6)[d] |
| Increase in reducing sugar production (%)[c] | 27.2[e] | |

Notes.
[a] 37 $\mu$g or 0.0067 U of CBHB was added.
[b] 2,996 $\mu$g or 0.36 FPU of Cellic® CTec2 was added.
[c] Total reducing sugar (TRS) was calculated based on Eq. (1).
[d] Values predicted by RSM model is bracketed.
[e] $P < 0.005$; a significant increase from (b) to (c).

A calculation based on Eq. (4) indicated that maximum saccharification of OPEFB occurred when 33.8% Cellic® CTec2 was added. CBHB supplements were desirable as they further increased hydrolysis of OPEFB when the performance of Cellic® CTec2 was theoretically capped at an enzyme loading of 33.8% (producing only 17.2 mM of reducing sugars or 31.1% TRS).

Using the optimised enzyme loadings containing 29.96% Cellic® CTec2 and 0.37% CBHB, reducing sugar production was increased to 21.5 mM (or 38.8% TRS), equivalent to a 27.2% enhancement over the un-supplemented samples or a 25% increase over the theoretical maximum saccharification using Cellic® CTec2 alone (Table 5).

Use of FTIR has been proven to be one of the most useful methods for the characterisation of natural cellulose fibres. Furthermore, FTIR can provide researchers with further information on the supramolecular structure and the chemical compositions of cellulose fibres with minimal efforts in sample preparation (*Fan, Dai & Huang, 2012*). Following the addition of CBHB, the ATR-FTIR spectra of OPEFB samples shifted (Fig. 9) implying compositional changes as different functional groups absorb infra-red at other wavelengths (*Berthomieu & Hienerwadel, 2009*). Supplementation with CBHB significantly reduced peak intensities at 3,274 and 3,220 cm$^{-1}$ that are related to the presence of crystalline Cellulose I$\beta$ (3,274 cm$^{-1}$) and Cellulose I$\alpha$ (3,220 cm$^{-1}$) (*Popescu et al., 2013*) (another peak at 1,038 cm$^{-1}$ was unassigned, however this was based on information currently available). In other words, supplementing Cellic® CTec2 with CBHB had appreciably reduced crystalline cellulose content in OPEFB.

DNS assays showed that removal of crystalline cellulose by CBHB corresponded to an increase in reducing sugar levels as insoluble cellulose was converted into soluble sugars (Table 5). Since CBHB alone is inactive against crystalline cellulose, these data suggest that it might work synergistically with other components of Cellic® CTec2. Similar observations had been reported for hydrolysis of OPEFB by *Thielavia terrestris* CBH7B (*Woon et al., 2016b*) and degradation of cotton fibres by *P. decumbens* CBHI (*Gao et al., 2012*).

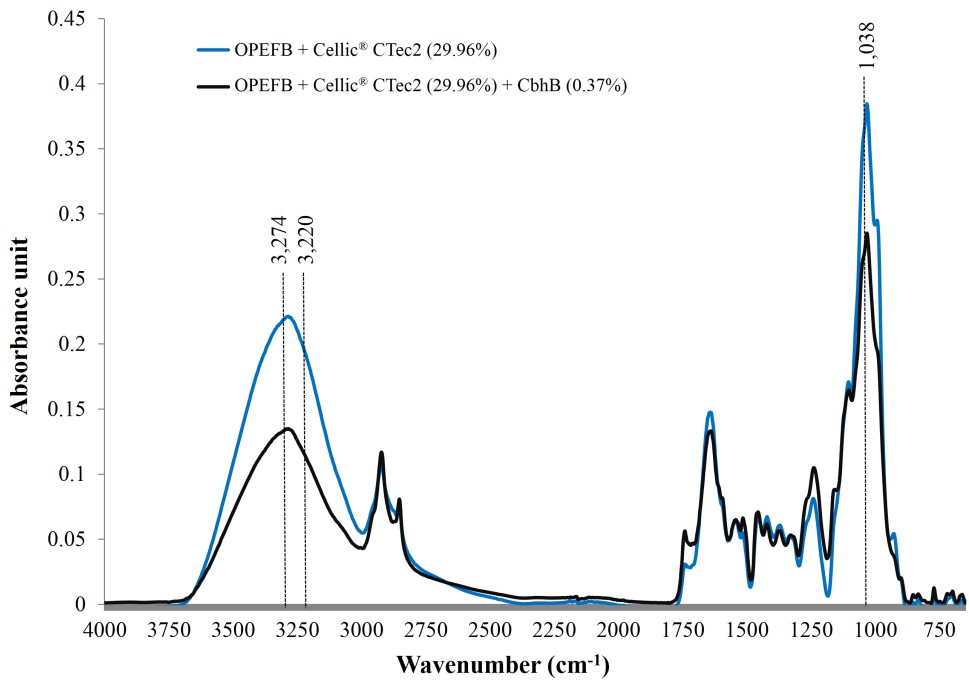

**Figure 9** **ATR-FTIR spectra of OPEFB treated with Cellic® CTec2 and Cellic® CTec2 supplemented with CbhB.** Dried OPEFB samples were scanned from wavenumber of 650 to 4,000 cm$^{-1}$ using a Perkin Elmer Spectrum 400 series equipped with Universal ATR sampler accessories.

# CONCLUSIONS

Cellobiohydrolase B from *A. niger* was over-expressed in *P. pastoris*. The recombinant CBHB functioned optimally at 50 °C and pH 4. This enzyme, originated from a known mesophilic fungus, displayed moderate thermal stability as it retained more than 80% residual activity upon incubation at 80 °C for 30 min. The kinetic constants $K_m$ and $V_{max}$ of CBHB towards MUC substrate were 0.25 mM and 1.41 U mg$^{-1}$, respectively. Supplementation of the commercial cellulase cocktail Cellic® CTec2 with this enzyme boosted saccharification of OPEFB by 27%. Recombinant CBHB from *A. niger* is thus a useful supplement towards commercial enzyme cocktails such as Cellic® CTec2 in converting OPEFB to simple sugars.

# ACKNOWLEDGEMENTS

The authors acknowledge the Malaysian Genome Institute (MGI) and the Centre for Research and Instrumentation Management (CRIM) at the Universiti Kebangsaan Malaysia for technical support.

### Funding

This work was supported by the Malaysian Ministry of Science, Technology and Innovation under research grant 10-05-MGI-GMB001/2 and TF0310F086. This work was also supported by the MyBrain15 scholarship, which was granted to James Woon Sy-Keen. The funders had no role in study design, data collection and analysis, decision to publish, or preparation of the manuscript.

### Grant Disclosures

The following grant information was disclosed by the authors:
Malaysian Ministry of Science, Technology and Innovation: 10-05-MGI-GMB001/2, TF0310F086.
MyBrain15.

### Competing Interests

The authors declare there are no competing interests.

### Author Contributions

- James Sy-Keen Woon conceived and designed the experiments, performed the experiments, analyzed the data, wrote the paper, prepared figures and/or tables.
- Mukram M. Mackeen and Abdul Munir Abdul Murad conceived and designed the experiments, analyzed the data, contributed reagents/materials/analysis tools.
- Rosli M. Illias and Nor M. Mahadi contributed reagents/materials/analysis tools.
- William J. Broughton analyzed the data, wrote the paper, reviewed drafts of the paper.
- Farah Diba Abu Bakar conceived and designed the experiments, analyzed the data, contributed reagents/materials/analysis tools, wrote the paper, reviewed drafts of the paper.

### Data Availability

    The raw data is included in the Results and Discussion sections of the manuscript and in two Supplemental Files.

### Supplemental Information

Supplemental information for this article can be found online at http://dx.doi.org/10.7717/peerj.3909#supplemental-information.

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
