# Peer review of "Cellobiohydrolase B of Aspergillus niger over-expressed in Pichia pastoris stimulates hydrolysis of oil palm empty fruit bunches"

_PeerJ, doi:10.7717/peerj.3909_

## Round 0.1 · original submission · Major Revisions

The peer-review reports from your work are positive, but additional detail is needed before I can accept your manuscript. Please address the helpful comments provided by our reviewers.

Additional review-level comments by the Editor:

Present results should be compared to those in the 2012 paper "Cloning of a cellobiohydrolase gene (cbh1) from Aspergillus niger and heterogenous expression in Pichia pastoris" (DOI:10.4028/www.scientific.net/AMR.347-353.2443), which should be cited as a previous expression of the title enzyme in the same model organism.

p.65 IMAC abbreviation is not defined.

p.178
How did you determine that Na2CO3 effectively terminated the reaction?

throughout "Material and Methods" section, please include amount of enzyme used in each experiment.

Table 2: please use substrates full names, rather than abbreviations. pH and temperature should be shown in the legend or title .

Table 3: Rows seem to be arranged randomly. Please re-order them in some consistent way to facilitate visual analysis of underlying trends.

Table 5: Please state amounts of enzyme in mg and U instead of % in situations a) b) and c).

The legend to Figure 6 seems to contain a mistake, since the concentrations read from the graph do not correspond to the concentrations stated in the legend.

fig. 8 please include the meaning of the contour lines.

Reviewer 1 ·

Basic reporting

The manuscript was well and clearly written (with very little grammatical errors), and was backed up by relevant literature references

Experimental design

Experimental design was suitable and robust for the research objectives spelt out.

Validity of the findings

Results and findings obtained were substantiated by raw data, and conclusions drawn were in accordance with the objectives outlined. However, results on biochemical characterization of the purified CbhB should also be highlighted in the conclusion as an answer to one of the objectives set out (line 115).

Additional comments

1. The English language can be improved, especially for lines 127 (growth of the yeast were performed), line 246 (incongruent sentence), line 273, 279, 296, 300, line 336- mentioned, not mention, line 353- Insensitivity of CbhB rather than CbhB insensitivity, line 370, line 379, line 382, line 384,
2. Please include a close bracket after mL-1, in line 144
3. line190: sugar produced, not product
4. Line 230- state the equipment used
5. Line 346- use of "of course" not quite appropriate
6. Line 353- was statistical test carried out? Cu and Ba seemed to affect at 10mM concentrations, as seen by the reduction of relative activity to less than 80%.
7. line 358- Gurgu and Barby, not Gurgu et al.
8. line 367- rephrase. The loadings were optimised, not the hydrolysis.
9. line 386- was, not is. Also line 391, 396
10. line 390. Use primary reference
11. line 416. please add a qualifier
12. References: please spell out all the authors's names to replace the et al. (lines 441, 466, 475, 481, 524
13. line 521: italicise genus and species names
14. Table 1, line 2- specific activity, not spesific
15. Table 2: table should be self-explanatory. Please write the names of the substrates in full in the notes below the table
16. Figure 4, Lane 2. IMAC- purified CbhB deglycosylated using PNGaseF, rather than deglycosylated IMAC- purified CbhB using PNGaseF
17. Figure 8. Range of Cellic CTec2 in the title was 1.5-30%, but in the actual figure was 5.50 -25%

Reviewer 2 ·

Basic reporting

None

Experimental design

The procedures were efficient to produce and purify an Aspergilllus niger CBH in P. pastoris. Produced enzyme was properly evaluated and used as an additional enzyme during hydrolysis of a model substrate hydrolysis (pretreated oil palm empty bunch). Hydrolysis intervals in this substrate were shorter than usual (5h), but results were adequate for evaluating the recombinant enzyme mixed with commercial CellicCtec.

Experimental design was adequate

Validity of the findings

Experimental data are well interpreted and discussed

Additional comments

The work is a general contribution to the area related to enzymes used in biomass conversion. Production and purification of a recombinant CBH was developed.
Over glycosylation of the recombinant enzyme could be one of the causes of non-action on complex and insoluble substrates such as crystalline cellulose.

In general, the results are sound and merit publication

Reviewer 3 ·

Basic reporting

The manuscript by Woon et al., describes the heterologous expression and characterization of a CBH from A.niger, as well as its ability to increase the saccharification yields of oil palm empty fruit bunches when added to a commercial preparation.

The manuscript is well-written and literature references and scientific background are provided.

Experimental design

This study is within the Aims and Scope of the journal.

In general, the experiments are described with sufficient detail, were performed correctly and the conclusions follow the results.

Some parts of the experiments and results needs more clarifications, such as the cloning of the cbh2 gene (PCR amplification, primers and polymerase used) and the experimental design by response surface methodology (enzyme supplementation, choice of upper and lower limits).

Validity of the findings

In general, conclusions are well stated and supported.

There are some parts that need clarification, such as the low recovery yields of CBH2 and the imprortance of ATR-FTIR data on evaluation of CellicCtec2 supplementation with CBH2 and the changes on substrate morphology.

Additional comments

- Lines 62-63: Keywords need to be more specific and representative of this work, for example bioethanol and lignin are not within the scope of this study.
- Line 81: “Aspergillus niger is a filamentous fungus well-known for its ability to produce copious amount of lignocellulose degrading enzymes”: this paper by Zoglowek et al., 2015 mainly describes how A.niger has been used as a host for heterologous production of CBHs and other cellulases.
- Line 89: T.reesei has low extracellular β-gucosidase activity, but this fungus expresses many intracellular and cell-bound CBHs.
- Lines 90-92: Supplementation is also used because different types of biomass have different properties, therefore they require a set of enzymes that has to be tailor-made. For example, endoglucanases are needed for amorphous substrates, whereas CBH supplementation is preferred for substrates with high crystallinity.
- Lines 93-95: cellulases consortium has been reported much earlier than 2016, so another reference is suggested. The same for line 103, for CBHs acting on reducing and non-reducing cellulose chain end.
- Line 106-107: Apart from glycosylation, there are other post-translational modifications that affect the enzyme performance and activity, ex. N-terminal pyroglutamate fo. rmation that is commonly observed in crystal structures of Cel7A enzymes and is related to enzyme stability, is not possible to occur in the Pichia pastoris expression system. This makes the recombinant enzymes less active and less stable than their native counterparts (Biotechnol. Bioeng. 2014;111: 842–847).
- Line 130-131: A full description of the cloning procedure is missing, i.e. information about the gene (exons, introns), the primers that were used and the PCR conditions, as well as the polymerase used. This part is also missing from the Results and Discussion Section (Line 262-263), where cbh gene should be analyzed for glycosylation sites, signal peptide and homology with characterized CBHs and an accession number for Genbank should be given. There is some information on Lines 295-305, but this part should be re-written again carefully. Line 302, “molecular mass of CbhB to ~ 70 to 100 kDa”: it is not clear what authors mean.
- Line 170-172: After the digestion with PNGaseF, how was the extent of N-glycosylation estimated? Did authors perform an SDS to compare the molecular weight of the enzyme before and after the deglycosylation? If so, this should be mentioned here.
- Lines 197-198: Did authors checked the stability/autohydrolysis of MUF at high temperatures, such as 70-80οC? Did they run control reactions without the addition of enzyme?
- Line 206: What was the CBH concentration that was used for calculating the kinetic parameters? How was the end product inhibition estimated?
- Line 217: Authors should choose another title, more specific than “complementary effects”. Same for Line 366.
- Line 217-220: When a monoenzyme (for example a CBH) is added to a commercial mixture, then there are two types: one is when the enzyme loading of the commercial preparation remains the same and CBH is added, so the final enzyme loading increases (this is called addition/ supplementation), and the other one is when one part of the commercial preparation is replaced by the CBH and the final enzyme loading is constant (this is called replacement/complementation). It should be mentioned which type is being used here. 0
- Line 226: Did authors check the composition of the material (glucose, xylose, lignin) before and after the pretreatment?
- Line 233-235: Apart from %TRS rate, the %hydrolysis/saccharification rate could be also estimated by taking into account the glucose and/or xylose content of OPEFB according to the analysis of solid fiber fraction from the following equation (WIREs Energy Environ. 2013; 2, 633-654):
% hydrolysis rate=[TRS(mg/mL)/DM(mg/mL)*glucose content*1.111] *100
Did authors check glucose or xylose yields after enzymatic reactions?
- Line 245-247: How did the authors chose the limits of the enzyme and Cellic®CTec2 relative abundances? In general, it should be avoided not only working within a wide domain, as this may impact the reliability of predictions, but also limiting in a narrow domain, since extrapolation outside the borders is impossible. The lower and upper limits of each component should be chosen by combining data from the literature with rational consideration.
- Line 284-285 and Table 2 data: it is obvious that the enzyme is quite unstable and this can be attributed to many reasons, including the absence of post-translational modifications, such as the pyroglutamate formation ect. CBH is degraded and gradually loses its His-tag and this is the reason why it is not possible to be purified by IMAC and the purification fold decreases. Authors should give an explanation. Did they perform an SDS during the cultivation in order to check whether the molecular weight of the protein remains stable? What is there on the lowest part of the SDS-PAGE shown at Figure 2 (it is cut and not shown)?
- In order to check whether the addition of CBH reduced the substrate crystallinity or not, study of the X-ray diffraction spectra could give more accurate results than ATR-FTIR.
- According to gene/protein nomenclature, genes are denoted by three italicized lower case letters, while Protein names are the same as the gene names, but the protein names are not italicized, and the first letter is upper-case (see "Guidelines for Formatting Gene and Protein Names", http://www.biosciencewriters.com/Guidelines-for-Formatting-Gene-and-Protein-Names.aspx), so it should be cbhB gene and CBHB protein.
- The authors should also check carefully for references' format, as well as typing/editing and punctuation errors in the text and in the figure captions. All organism names should be written in italics.

---

## Round 0.2 · accepted · Accept

Thank yo for your thorough response to the reviewer's comments. I am glad to accept your paper for publication.

Reviewer 3 ·

Basic reporting

no comment

Experimental design

no comment

Validity of the findings

no comments

Additional comments

The manuscript was greatly imporoved after revision and now is accepted for publication in PeerJ.